# Urban Green Spaces and Vector-Borne Disease Risk in Africa: The Case of an Unclean Forested Park in Libreville (Gabon, Central Africa)

**DOI:** 10.3390/ijerph20105774

**Published:** 2023-05-10

**Authors:** Judicaël Obame-Nkoghe, Boris Kevin Makanga, Sylvie Brizard Zongo, Aubin Armel Koumba, Prune Komba, Neil-Michel Longo-Pendy, Franck Mounioko, Rodolphe Akone-Ella, Lynda Chancelya Nkoghe-Nkoghe, Marc-Flaubert Ngangue-Salamba, Patrick Yangari, Sophie Aboughe-Angone, Florence Fournet, Pierre Kengne, Christophe Paupy

**Affiliations:** 1Laboratoire de Biologie Moléculaire et Cellulaire, Département de Biologie, Université des Sciences et Techniques de Masuku (USTM), Franceville BP 941, Gabon; 2Unité de Recherche en Écologie de la Santé, Centre Interdisciplinaire de Recherches Médicales de Franceville (CIRMF), Franceville BP 769, Gabon; 3Institut de Recherche en Écologie Tropicale (IRET/CENAREST), Libreville BP 13354, Gabon; 4Département Faune et Aires Protégées, École Nationale des Eaux et Forêts (ENEF), Libreville BP 3960, Gabon; 5Unité de Recherche GéoHydrosystèmes Continentaux (UR GéHCo), Département Géosciences et Environnement, Université de Tours, 37000 Tours, France; 6Institut de Pharmacopée et de Médecine Traditionnelle (IPHAMETRA), Libreville BP 1156, Gabon; 7Unité Mixte de Recherche Maladies Infectieuses et Vecteurs, Écologie, Génétique, Évolution et Contrôle (MIVEGEC), Université de Montpellier, Centre National de la Recherche Scientifique (CNRS), Institut de Recherche pour le Développement (IRD), 34193 Montpellier, France

**Keywords:** vector-borne diseases, urbanization, urban forest, *Aedes*, *Anopheles*, *Culex*, Gabon, central Africa

## Abstract

In Africa, vector-borne diseases are a major public health issue, especially in cities. Urban greening is increasingly considered to promote inhabitants’ well-being. However, the impact of urban green spaces on vector risk remains poorly investigated, particularly urban forests in poor hygienic conditions. Therefore, using larval sampling and human landing catches, this study investigated the mosquito diversity and the vector risk in a forest patch and its inhabited surroundings in Libreville, Gabon, central Africa. Among the 104 water containers explored, 94 (90.4%) were artificial (gutters, used tires, plastic bottles) and 10 (9.6%) were natural (puddles, streams, tree holes). In total, 770 mosquitoes belonging to 14 species were collected from such water containers (73.1% outside the forested area). The mosquito community was dominated by *Aedes albopictus* (33.5%), *Culex quinquefasciatus* (30.4%), and *Lutzia tigripes* (16.5%). Although mosquito diversity was almost double outside compared to inside the forest (Shannon diversity index: 1.3 vs. 0.7, respectively), the species relative abundance (Morisita–Horn index = 0.7) was similar. *Ae. albopictus* (86.1%) was the most aggressive species, putting people at risk of *Aedes*-borne viruses. This study highlights the importance of waste pollution in urban forested ecosystems as a potential driver of mosquito-borne diseases.

## 1. Introduction

Vector-borne diseases (VBDs) account for a significant proportion of human diseases worldwide [1]. Globally, Africa is one of the most affected continents and malaria, arboviral diseases (e.g., yellow fever, chikungunya, dengue, Zika), and neglected tropical diseases (e.g., lymphatic filariasis) are the cause of several major health crises [1,2,3,4,5]. Among the affected regions, urban localities see a significant concentration of the VBD burden [6]. In central Africa, urban areas are commonly associated with a significantly higher prevalence of malaria [7,8,9,10,11] and arboviral diseases [12,13,14,15].

In central Africa, as in other sub-Saharan African countries, urbanization is poorly controlled. Insufficient wastewater and waste management have led to the proliferation of mosquito breeding habitats that put the population at risk of VBDs [16,17]. Therefore, urban planning and environmental management are important issues in African cities that must be addressed to mitigate the VBD risk.

Urban greening (i.e., promoting and developing green spaces, such as parks, agriculture areas, and ecological corridors, within cities) is a concept that has been promoted because of the potential benefits to the environment quality and human well-being [18]. Urban green spaces are also recognized as sustainable solutions to mitigate global warming, especially at the microclimatic scale, by regulating the ambient temperature in cities through the freshening provided by the tree shade [18]. In addition, the development of urban green spaces is valued for its positive effect on carbon sequestration and biodiversity conservation [19]. However, the potential downside of urban greening concerning public health, including VBDs, is not well understood. In temperate zones, some studies have shown that urban woodland vegetation cover facilitates dispersal and creates movement corridors for female *Aedes* mosquitoes in search of egg-laying sites, whereas grasslands with few tall grasses seems to limit them [20]. In tropical America, the presence of high vegetation might define a microclimate that locally influences the relative air humidity, leading to a positive association with the presence of *Aedes* mosquitoes [21]. Similarly, it has been shown that urban forests affect mosquito communities, including the two most important arbovirus vectors (*Aedes albopictus* and *Aedes aegypti*). These species, especially *Ae. albopictus,* may use the forest as a refuge and act as “bridge vectors” of arboviruses between the forest and anthropogenic settings [22]. However, in Africa, no study has investigated how urban green spaces shape mosquito communities and influence the related VBD risk.

In Gabon, 85% of the national population lives in urban areas [23], where VBD prevalence is the highest [12,24,25,26]. In Gabon, dense forest covers more than 80% of the national territory, placing the country at the upper end of the rate of forest area per capita in Africa [27]. Therefore, the question of the impact of urban green spaces on the modulation of VBD epidemiological patterns must be addressed. Specifically, it is not known how anthropogenic disturbances of such urban green spaces (e.g., littering) may influence the distribution, abundance, and diversity of mosquito vectors and the associated VBD risk for the surrounding residents. In the present study, we explored the mosquito diversity and larval microhabitat typology at the interface between a frequently visited forested reserve, which is used as dumping ground, and the bordering human habitat to evaluate the VBD entomological risk related to this ecosystem within the city of Libreville, Gabon’s capital, central Africa.

## 2. Materials and Methods

### 2.1. Study Area

This study was carried out from 14 July to 20 August 2020, during the long dry season (which lasts from June to September), in the Sibang arboretum (0°24′58″ N and 9°29′23″ E) and its inhabited surroundings (Figure 1). The study was carried out during the dry season because we suspected that this wooded area may serve as a refuge for mosquitoes during the period of low rainfall, and we wanted to identify the mosquito communities and the vector risk associated with this ecosystem. The Sibang arboretum is a forest park that covers 160,000 m^2^. It is located in an urbanized area of the eastern part of Libreville and is crossed by the Adoung river. Cordier (2000) recorded 137 plant species and at least 40 bird species, and a poorly quantified diversity of vertebrate animals including reptiles and small mammals (e.g., squirrels) [28]. Despite its protected status, the Sibang arboretum is subject to an important anthropogenic pressure, as indicated by the presence of diverse traces of human frequentation and the accumulation of rubbish at some locations up to several tens of meters inside the arboretum, especially along one major border separating the arboretum from the surrounding households (Figure 2).

### 2.2. Larval Sampling

Larval samplings were carried out during 13 non-consecutive days outside (over a radius of 800 m around the forest) and inside the forest. Mosquito breeding sites were investigated as exhaustively as possible, according to the field accessibility and permission from residents to visit their properties. During the collection period, the sampling effort lasted 15 h and 63 h inside and outside the forest, respectively.

Water containers were explored at the different sites (Figure 1C) in order to collect larvae and pupae using a dipper or a pipette and transfer them into vials labeled according to the container type, site location, and date. At the entomological laboratory of the Research Institute for Tropical Ecology (IRET), Libreville, larvae and pupae were placed into labeled trays covered with a mosquito net and maintained at room temperature until the emergence of adults. All samples were treated in the same conditions, from collection to rearing, to minimize the bias in abundance and diversity among sites. Upon emergence, adult mosquitoes were kept at −20 °C for 30 min to be euthanized, and then morphologically identified (species or genus) using a binocular microscope (Leica Microsystems©) and “customized” taxonomic keys based on the updates of the Edwards’ identification keys for Ethiopian mosquitoes [29], and the Huang’s key for the subgenus *Stegomyia* of *Aedes* mosquitoes from the Afrotropical region [30]. Species were named according to the online list of valid species (http://mosquito-taxonomic-inventory.info, accessed on 20 July 2020).

### 2.3. Adult Mosquito Collection

Adult female mosquitoes were collected using the human landing catch (HLC) technique during the daytime. The study was approved by the Gabon Ethics Committee (permit No. 016/2019/PR/SG/CNE). Three volunteers, posted at three fixed capture sites, collected adult females during two consecutive sampling sessions, one inside and one outside the forest. At each session, the three volunteers were separated by at least 50 m (Figure 1C). Each sampling session was for three consecutive days, from 10:00 a.m. to 2:00 p.m. (4 h per day), representing a sampling effort of 12 h. Mosquitoes were captured with a mouth aspirator upon landing on the volunteer’s bare legs and then transferred into a plastic jar covered with a net to prevent their escape. At the end of the day, mosquitoes were transported to the IRET entomological laboratory for identification, as described above.

### 2.4. Data Analysis 

All statistical analyses were carried out using the R software v3.6.1 (https://www.r-project.org/, accessed on 1 September 2020). Spatial analyses were performed using Quantum GIS version 3.10.7 (https://www.qgis.org/, accessed on 15 September 2020). Species richness was determined as the number of mosquito species recovered. Species diversity (i.e., number of species and their abundance) was assessed using the Shannon–Weaver index (*H*) [31] and the “*diversity*” function of the *vegan* package. To investigate the similarity in terms of species composition and the density between mosquito communities inside and outside the forest, the Morisita–Horn similarity index (*C*) [32] was calculated using the “*vegdist*” function of the *vegan* package. Because “*vegdist*” is an analysis of dissimilarity (*C′*), *C* = *1* − *C′* was used for this study. *C* ranged from 0 (0% of similarity between compartments) to 1 (100% of identity between compartments).

Environmental variables were collected to characterize the larval habitats exploited by mosquitoes in the Sibang arboretum and its surroundings. These variables included the substrate physical description, the type (artificial vs. natural), and the spatial location of larval habitats (inside vs. outside the forest). Multiple Correspondence Analysis (MCA) was used to assess the similarity of larval habitats according to the species composition and environmental characteristics. It allowed the assessment of the mosquito species’ degree of specificity related to the larval habitat type and location. The MCA was also used to identify the most relevant biotic and environmental variables associated with the larval habitat segregation.

A k-means analysis based on Ward’s method was performed using the *fpc* package [33] and the Calinski Harabasz index (CH index) [34] to determine the minimal parsimonious number of ecological species clusters. Based on the results of the Principal Component Analysis (PCA) using the *FactoMineR* package [35], Hierarchical Agglomerative Clustering (HAC) was used to determine and visualize species clusters within the same microecological niche.

The Wilcoxon’s test based on the HLC data was used to assess the aggressiveness of bloodmeal-seeking female mosquitoes according to the sampling location (inside vs. outside the forest). An analysis of variance (ANOVA) was performed to determine the differences in the number of captured specimens per person among species in each sampling location.

## 3. Results

### 3.1. Typology and Positivity of Larval Habitats Inside and Outside the Forest

In total, 104 water containers were investigated in the Sibang district (Figure 1): 28 (26.9%) inside (8 natural and 20 artificial) and 76 (73.1%) outside the forest area (2 natural and 74 artificial) (Table 1). However, the number of water containers recovered per hour was 1.9 inside the forest and 1.1 outside the forest. Overall, 10 water containers (9.6%) were natural and 94 (90.4%) were artificial (Figure 3A). The natural water containers were tree holes (50%), streams (40%), and a puddle (10%). Artificial water containers included plastic containers (34%), puddles (19.1%) (caused by water leakage from a piped system), gutters (18.1%), worn tires (14.9%), and other kinds of waste (<15%), including a discarded freezer, a tin can, a glass jar, a wash basin, and various metallic containers (Table 1). Almost all water containers (*n* = 27, 96.4%) inside the forest and all water containers (*n* = 76, 100%) outside the forest were positive (i.e., presence of at least one mosquito larva or pupa in the water container) (Figure 3B). Inside the forest, all natural (*n* = 8, 100%) and almost all artificial (*n* = 19, 95%) containers were positive. Outside the forest, all natural (*n* = 2, 100%) and artificial (*n* = 74, 100%) containers were positive.

### 3.2. Mosquito Species Composition and Diversity

After larval rearing at the insectary, 770 adult mosquitoes emerged and were morphologically identified. These mosquitoes belonged to fourteen species grouped into six genera, including *Aedes* (two species), *Culex* (eight species and one undetermined), *Anopheles* (one species), *Eretmapodites* (one species), *Lutzia* (one species), and *Toxorhynchites* (one species) (Table 2). Overall, the species assemblage was largely dominated by *Ae. albopictus* (33.5%), *Culex quinquefasciatus* (30.4%), and *Lutzia tigripes* (16.5%), whereas *Culex cinerellus*, *Culex decens*, *Culex duttoni, Culex trifilatus*, *Culex umbripes*, *Culex univittatus*, *Eretmapodites inornatus*, and *Toxorhynchites evansae* were relatively less abundant (<1% of the total assemblage for each species) (Table 2). Inside the forest, *Ae. albopictus* (77%), *Cx. quinquefasciatus* (13.8%), and *Lu. tigripes* (6.9%) predominated in artificial habitats, whereas *Culex tauffliebi* (50%) and *Lu. tigripes* (22.7%) were the most predominant species in natural habitats (Table 2). Outside the forest, *Cx. quinquefasciatus* (35.1%), *Ae. albopictus* (29.7%), and *Lu. tigripes* (17.1%) were the most predominant species in artificial containers. *Cx. tauffliebi* (67.8%) and *Lu. tigripes* (28.6%) were the most predominant species recovered in natural water containers. *Ae. albopictus* and *An. gambiae* s. l. were absent from all natural habitats investigated outside the forest (Table 2). The Shannon–Weaver index indicated that mosquito species diversity was lower inside than outside the forest (H = 0.7 vs. 1.3). However, overall, the community similarity level was comparable in both compartments (C = 0.7; i.e., 70% of similarity, which was associated with the amount of shared species between compartments and their relative abundance within the respective communities). 

### 3.3. Larval Habitat Typology, Similarity and Species Clustering

*Ae. albopictus* was mainly found in artificial breeding containers. Additionally, this species was associated with a higher relative abundances in both compartments compared with the other species (Table 2). Similarly, *Cx. quinquefasciatus* was mostly recovered in artificial breeding containers (13.8% and 35.1% inside and outside the forest, respectively). Conversely, *Lu. tigripes* was relatively more abundant in natural containers (22.7% and 28.6% inside and outside the forest, respectively) (Table 2). *Ae. aegypti* was only recovered in artificial breeding containers and outside the forest. *An. gambiae* s. l. was almost exclusively found in artificial breeding sites outside the forest (6.5%) (Table 2).

The MCA revealed that the best-correlated variables associated with the larval habitat distribution were the habitat type (natural or artificial) and spatial location (inside or outside the forest), and also the presence of the species *Ae. albopictus*, *An. gambiae* s. l., *Ae. aegypti*, *Lu. tigripes*, *Cx. quinquefasciatus*, *Cx. trifilatus*, *Cx. duttoni*, and *Er. inornatus* (Appendix A). These variables explained 38.4% of the total variance associated with the species composition of larval habitats. This analysis showed a clear segregation of larval habitats according to their spatial location and type (Figure 4A,B). In terms of larval habitat specificity, our results revealed that mosquito larval habitats were likely to be exploited by species that could be characterized as specialist (i.e., with a high level of habitat specificity: *Ae. aegypti*, *An. gambiae* s. l., *Cx. trifilatus*, *Cx. duttoni*, and *Er. inornatus*), opportunistic (i.e., species that, unlike specialist species, can adapt to a range of environmental conditions: *Ae. albopictus* and *Lu. tigripes*) or ubiquitous (i.e., species highly adapted to occupy and proliferate in varied ecological niches, possibly with a wide geographical distribution: *Cx. quinquefasciatus*) (Figure 4C–J). The most exploited breeding sites by *Ae. albopictus* (opportunistic species) and by *Cx. quinquefasciatus* (ubiquitous species) were discarded plastic containers and worn tires/metallic or concrete containers (i.e., wash basin, see Table 1), respectively (Figure 5). *Lu. tigripes* (opportunistic species) used discarded tires, puddles, plastic, metallic or concrete containers (Figure 5). *Ae. aegypti* and *An. gambiae* s. l. (the main vectors of public health concern) exclusively exploited plastic containers and puddles (both natural and artificial), respectively, as breeding sites (Figure 5).

To determine the degree of ecological niche similarity among mosquito species at the larval microhabitat scale, the k-means method based on the PCA (72.8% of the total explained inertia of count data over the first two principal components, see Appendix A) and the CH index revealed that the minimal parsimonious number of ecological clusters of species was four (Figure 6A). Two of these clusters were mono-specific (cluster 2: *Ae. albopictus*; cluster 3: *Cx. quinquefasciatus*) and two were multi-specific (cluster 1: *Ae. aegypti*, *Tx. evansae*, *Cx. umbripes*, *Cx. decens*, *Cx. trifilatus*, *Cx. cinerellus*, *Cx. univittatus*, *Cx. duttoni*, *Culex* sp., *Er. inornatus*; cluster 4: *Cx. tauffliebi* and *Lu. tigripes*) (Figure 6B). Our analysis showed that species in the same cluster were more likely to share the same type of larval ecological niche.

### 3.4. Biting Patterns of Mosquito Species

In total, 874 female mosquitoes from three species were captured and identified, 755 (86.4%) inside and 119 (13.6%) outside the forest. *Ae. albopictus* (*n* = 753 specimens captured; 86.1%) was the main aggressive species for humans, with a peak of aggressiveness of 35.2 bites/person/hour (bph) reached between 11:00 a.m. and 12:00 p.m. inside the forest (Figure 7A). *Ae. aegypti* (*n* = 108, 12.4%) and *Er. inornatus* (*n* = 13, 1.5%) specimens were less aggressive (<5 bph regardless of the spatial location). The ANOVA test showed that overall, *Ae. albopictus* was significantly more aggressive than *Ae. aegypti* and *Er. inornatus* (F = 11.8; df = 2; *p* < 0.001). Moreover, *Ae. albopictus* was significantly more aggressive inside than outside the forest (W = 140, *p* < 0.001) (Figure 7B), whereas no significant difference between compartments was found for *Ae. aegypti* (W = 84.5, *p* = 0.5) (Figure 7B). *Er. inornatus* was only captured inside the forest (Figure 7B).

## 4. Discussion

### 4.1. Mosquito Communities in the Urban Forested Area of Sibang

This entomological survey described the species diversity and the larval microhabitat typology of mosquito communities in an urban forested reserve and in its direct surroundings (within the city of Libreville, Gabon) to evaluate whether this environment full of waste influenced the VBD risk. Most of the surveyed potential breeding sites (≥95%) inside and outside the forest area (i.e., tires, plastic containers, gutters, tree holes) contained mosquito larvae. The predominant mosquito genera were *Aedes*, *Culex*, and *Lutzia,* similar to what has previously been reported in previous investigations in Gabon (*Aedes*, *Culex*, and *Lutzia* mosquitoes) [36,37,38], central Africa [39,40,41], Asia [42,43], and South America (*Aedes* and *Culex* mosquitoes) [44,45,46].

Overall, *Ae. albopictus*, *Cx. quinquefasciatus*, and *Lu. tigripes* were the most frequently found species. The strong presence of *Ae. albopictus* can be explained by the availability of artificial water containers that are suitable habitats for this species [40]. Previous studies demonstrated *Lu. tigripes*’ natural predatory ability over other mosquito species, including mosquitoes of the *Aedes* genus [47,48,49,50]. Thus, its abundance and distribution could follow the dynamics of the other species used as prey. The relatively high abundance of *Cx. quinquefasciatus* confirms its ubiquity in various habitats, especially artificial habitats mostly found in highly populated areas, such as urban settings [39]. *An. gambiae* s. l. (the major malaria vector in the world) was almost entirely found outside the forest in open puddles due to domestic wastewater runoff (a typical larval habitat created by human activity for this species) [51,52,53,54], and also in a discarded tire. This confirms the use of unusual microhabitats, including those exploited by *Aedes* mosquitoes, as described for *Anopheles stephensi*, an emerging malaria vector in Africa [55].

Although some sites, such as epiphytic plants, tree holes at high elevations, and underground animal burrows, were not explored because of access difficulties, our observations indicated that the mosquito diversity within the forested compartment was two-fold lower than that outside the forest. Moreover, in the forested compartment, no species presumed to be exclusively sylvatic was detected, whereas some species already observed in a natural sylvatic condition elsewhere in Africa were identified. Indeed, Diallo et al. [56] observed *An. gambiae* s. l., *Cx. decens*, and *Cx. quinquefasciatus* in a forest canopy in Senegal (Kédougou region), although these three species were rare in that forested habitat. Similarly, Pereira dos Santos et al. [57] found *Ae. albopictus* up to several hundred meters inside an urban forest in Brazil. The absence of exclusively sylvatic species could be due to the lack of suitable conditions for sylvatic species, such as natural breeding sites (e.g., leaf axils, tree holes, rock holes, fruit shells), or the absence of animal host species for their blood meals. Extending larval sampling to the rainy season could have increased the number of detected species, including potential forest specialist species. Lastly, despite differences in mosquito diversity, the two compartments (i.e., inside and outside the forest) showed quite an important similarity during the dry season, mostly due to the high relative abundance of common species (i.e., *Ae. albopictus* and *Cx. quinquefasciatus*), thus rendering their communities similar.

Our analyses showed a clear segregation of larval habitats based on their type and spatial location. This could be explained by the inherent ecological preference of the recovered species. This suggests that in the Sibang district, there are only few mosquito species with a limited, specific ecological niche. For instance, *Ae. aegypti* was exclusively found in plastic artificial containers outside the forest, and clustered with other species including mainly *Culex* spp. Similarly, *An. gambiae* s. l. was almost exclusively found in ground puddles. On the other hand, *Ae. albopictus* and *Lu. tigripes* exploited artificial and natural microhabitats inside and outside the forest. These two opportunistic species, characterized by higher ecological plasticity, did not cluster together, possibly because of different microhabitat preference. Lastly, *Cx. quinquefasciatus* (a ubiquitous species with a large ecological niche) was recovered mostly in metallic and concrete containers, but also in tires and plastic containers.

### 4.2. The Mosquito Proliferation Drivers in the Urban Forested Area of Sibang

In the field, the larval infestation level of artificial water collections was very high, even inside the forest that is obviously used as a waste dump (Figure 2). This abundance of human-sourced breeding sites tends to indicate a convergence of breeding site types (mostly associated with waste dumping), available hosts, and consequently mosquito communities, both inside and at the forest periphery. The high level of anthropogenic pressure from neighboring households (accumulation of domestic waste inside the forest) promotes the proliferation of major disease vector species, such as *Ae. albopictus*, *Ae. aegypti*, and *Cx. quinquefasciatus*, that also breed in artificial water containers.

### 4.3. Mosquito Aggressiveness in the Urban Forested Area of Sibang

During the HLC-based collection time (10:00 a.m.–2:00 p.m.), *Ae. albopictus* was the predominant species collected, with a peak of aggressiveness between 11:00 a.m. and 12:00 p.m. and a biting rate of 35.2 bph inside the forest. Kamgang et al. [58] in Cameroon and Delatte et al. [59] in La Reunion reported peaks of aggressiveness for *Ae. albopictus* later during the daytime, between 4:00 p.m. and 5:30 p.m., suggesting a higher aggressiveness of this species in the Sibang area.

*Ae. albopictus* is a worldwide invasive arboviral vector of major public health concern [60] that has already caused chikungunya outbreaks in Gabon [12,61,62]. To the best of our knowledge, *Ae. albopictus* biting rates inside and outside a forested compartment, both in anthropized and wild environments, are not well documented. Nevertheless, *Ae. albopictus*’ aggressiveness level in this study was >2-fold higher than what had previously been observed in Libreville in suburban neighborhoods (15.7 bph) [37], some of which were wooded areas with chikungunya transmission records. Furthermore, in 2009, a study in the Central African Republic, which included forested peri-domestic areas among the sampling sites, reported a peak biting rate of 1.7 bph [63]. This low rate could be explained by the fact that this previous study was carried out during the early stage of the *Ae. albopictus* invasion in this country, when its density was still low. Alternatively, the sampled forested peri-domestic areas might not have been areas of waste dumping, which seems to be, based on our results, a driver of *Ae. albopictus* proliferation and aggressiveness. More surveys are needed to monitor the *Ae. albopictus* daytime biting rate over a longer period. In terms of public health, our results indicate the very high risk of diseases transmitted by *Ae. albopictus* for the human population in this area of Libreville, and the need for disease outbreak surveillance programs.

*Ae. aegypti* was the second most aggressive species, especially inside the forest. The low biting rate and the non-significant difference in the captured *Ae. aegypti* females between locations inside and outside the forest could be explained by its scarcity in the Sibang district. In addition, the relatively low proportion of *Ae. aegypti* larvae, compared with *Ae. albopictus*, suggests a population decline for this species due to the successful invasion of *Ae. albopictus*, as suggested in central Africa [37,41,64] and elsewhere in the world [65,66].

*Ae. albopictus* is well known for its opportunistic blood-feeding behavior and high vector competence for a number of disease-causing viruses [67]. Studies in the Sibang arboretum in the early 2000s reported a high biodiversity of vertebrate animals, including several species of small mammals, reptiles and birds [28]. Thus, the high density of *Ae. albopictus* might not only increase the risk of the inter-human transfer of *Ae. albopictus*-borne pathogens, but also represent a risk for potential zoonotic pathogens that could be hosted by the vertebrate animals in this forested patch, including birds and rodents, and be transmissible by this vector. Both animal groups are recognized hosts (or potential hosts) for zoonotic arboviruses, including West Nile virus for birds [68].

### 4.4. Mosquito Aggressiveness and Urban Greening

This study showed the important aggressiveness of *Ae. albopictus* and the risk of arbovirus transmission associated with this urbanized and forested district of Libreville. Such a risk could be exacerbated by the fact that this forested area might constitute a human-maintained incubator ecosystem and resting place for vectors, and therefore might facilitate and sustain the disease spread during epidemic periods.

Our results also highlighted that the poor sanitation of such an urban green area might modulate the burden of vector-borne diseases by intensifying (due to a high density of mosquitoes) or diluting (due to the presence of alternative hosts for mosquitoes to feed on) disease transmission. Araujo et al. [68] showed that in a Brazilian city, dengue incidence was higher in heat islands where vegetation cover was low than in forested neighborhoods considered to be fresher. Therefore, urban forests could be “islands of coolness” that might mitigate the spread of arboviruses, for example by slowing down the replication rate of viruses in mosquito vectors that live under the forest cover, as hypothesized in previous studies [69,70,71]. However, this positive effect could be counteracted by the level of pollution (due to poor sanitation), as shown by the present study. This requires effective sanitation measures in urban green spaces to mitigate the risk of VBDs in such areas. Thus, developing clean green spaces (e.g., in temperate poorly forested countries) and managing them with appropriate sanitation measures (e.g., in tropical and highly forested countries) could improve biodiversity, climate warming mitigation, and also the inhabitants’ well-being. Therefore, urban forest islands, such as the Sibang forest, should be well planned and managed to mitigate the risks of environmental-driven diseases in general, and VBDs in particular. 

### 4.5. Limitations of the Study

The results of this study are based on data collected during a 1.5-month dry season, and may not be generalizable to what might be observed during other seasons of the year or over a full annual climate cycle. Therefore, additional studies are needed to obtain a finer insight into the mosquito distribution, abundance, population dynamics, and diversity in this urban forest area of Gabon, as well as into the associated VBD risk.

Moreover, potentially important larval habitats, and thus a potentially important portion of mosquito diversity and abundance, might not have been investigated, particularly because of the limited access to properties due to uncooperative residents.

Due to government restrictions associated with the COVID-19 pandemic, HLC-based collections were made only during the day, and thus the mosquitoes collected were only day-biting species. Thus, the results do not take into account exclusively night-biting mosquito species, which may also be an important component of the vector risk associated with this urban forest. In addition, the HLC-based collections inside and outside the forest were not made concomitantly but sequentially, because the same three volunteers were always used to minimize the bias due to attractiveness variability among volunteers. However, no apparent major environmental variation (e.g., population movements, rainfall or other meteorological shifts) that could affect the collection outcomes was noticed.

## 5. Conclusions

This study in the urban forested area of Sibang allowed the recording of approximately 12% of the currently known mosquito species in Gabon [37,72,73,74,75]. The investigation revealed a variety of larval habitats exploited by the mosquito species in the study area. Most of these habitats were artificial and man-made. The study highlighted the importance of considering urban forested ecosystems, especially when associated with poor hygiene conditions, as potential drivers of disease emergence and spread in urban areas, which should be taken into account when designing vector control strategies. In Gabon, this study should contribute to guiding vector control strategies, particularly through the implementation of policies to ensure good environment management and the surveillance of vectors in urbanized areas.

## Figures and Tables

**Figure 1 ijerph-20-05774-f001:**
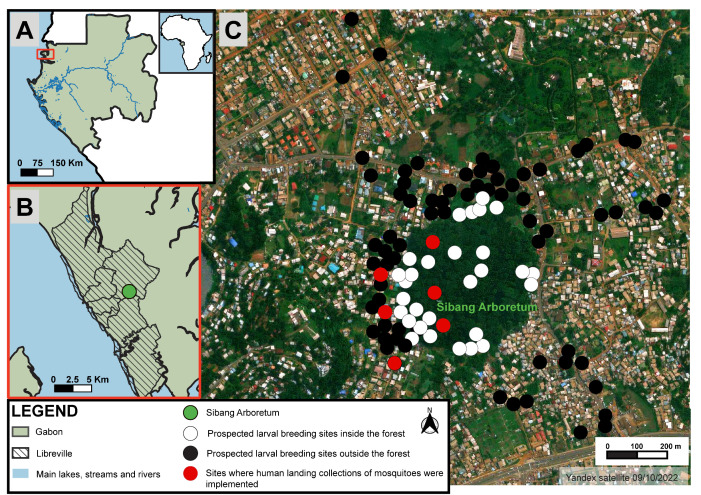
Study area and sampling sites. (**A**): Location of Gabon within Africa; (**B**): Location of the Sibang arboretum within Libreville; (**C**): Sampling sites. The locations of the spots were adjusted so that they do not overlap.

**Figure 2 ijerph-20-05774-f002:**
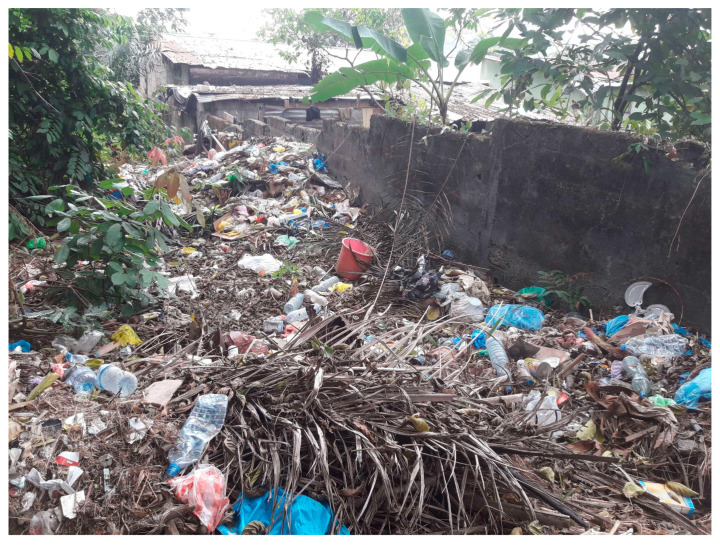
Example of waste dumping in the Sibang arboretum.

**Figure 3 ijerph-20-05774-f003:**
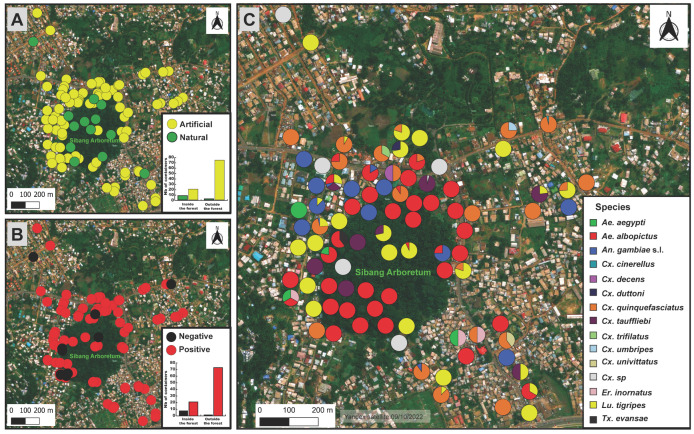
Spatial distribution of larval habitats and relative abundance of mosquito species in each larval habitat. (**A**): The yellow and green circles correspond to artificial and natural larval habitats, respectively. (**B**): Red and black circles represent positive (with at least one larva or pupa) and negative habitats, respectively. (**C**): Pie chart showing the species relative abundance at the different sampling locations.

**Figure 4 ijerph-20-05774-f004:**
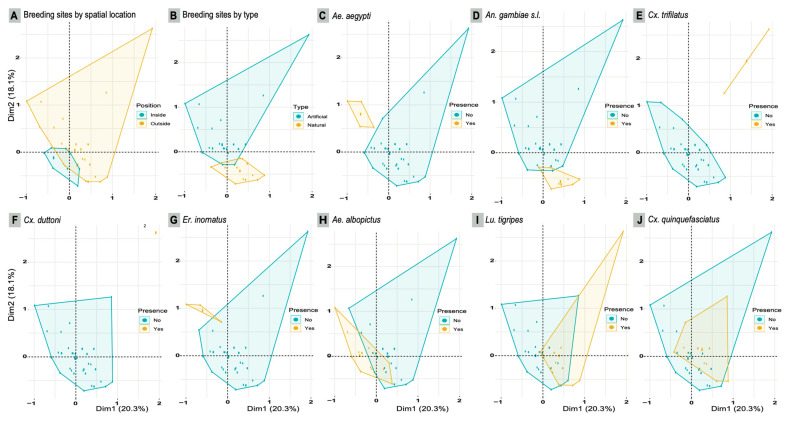
Distribution of larval habitats according to the retained variables processed in the MCA. The MCA explained 38.4% of the total variability of larval habitats in terms of species composition. These variables include spatial location (**A**) and type (**B**) of larval habitats, as well as the main species recovered (**C**–**J**). Intersection areas refer to habitats that tend to be similar in type (natural/artificial), location (inside/outside forest), and specific composition.

**Figure 5 ijerph-20-05774-f005:**
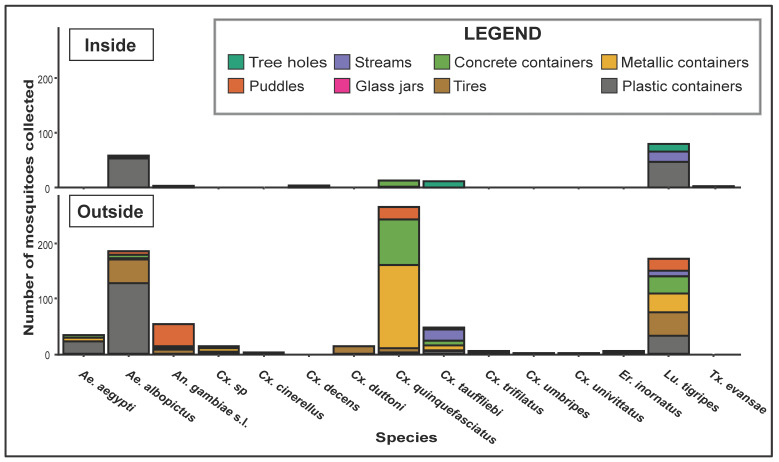
Mosquito species distribution based on the larval habitat nature. Concrete containers include the wash basin and gutters.

**Figure 6 ijerph-20-05774-f006:**
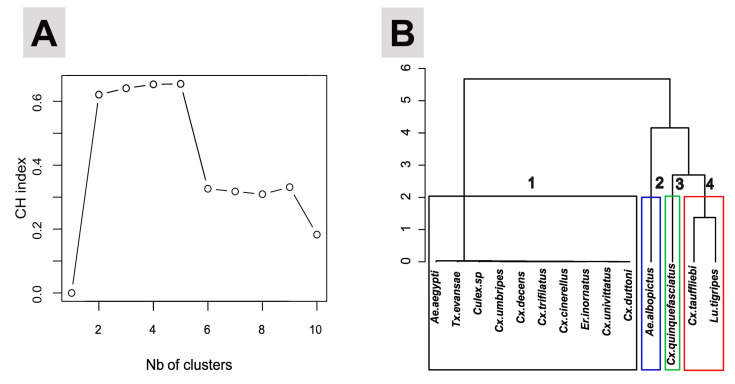
Species clustering based on the CH index and HAC, and analytical synopsis of the niche similarity level across mosquito species based on larval microhabitats. (**A**): Results of the Ward k-means analysis combined with the CH Index, revealing the parsimonious number of species clusters (*n* = 4). (**B**): Representation of the species clusters based on the HAC. Box colors refer to clusters, black (cluster 1), blue (cluster 2), green (cluster 3), red (cluster 4).

**Figure 7 ijerph-20-05774-f007:**
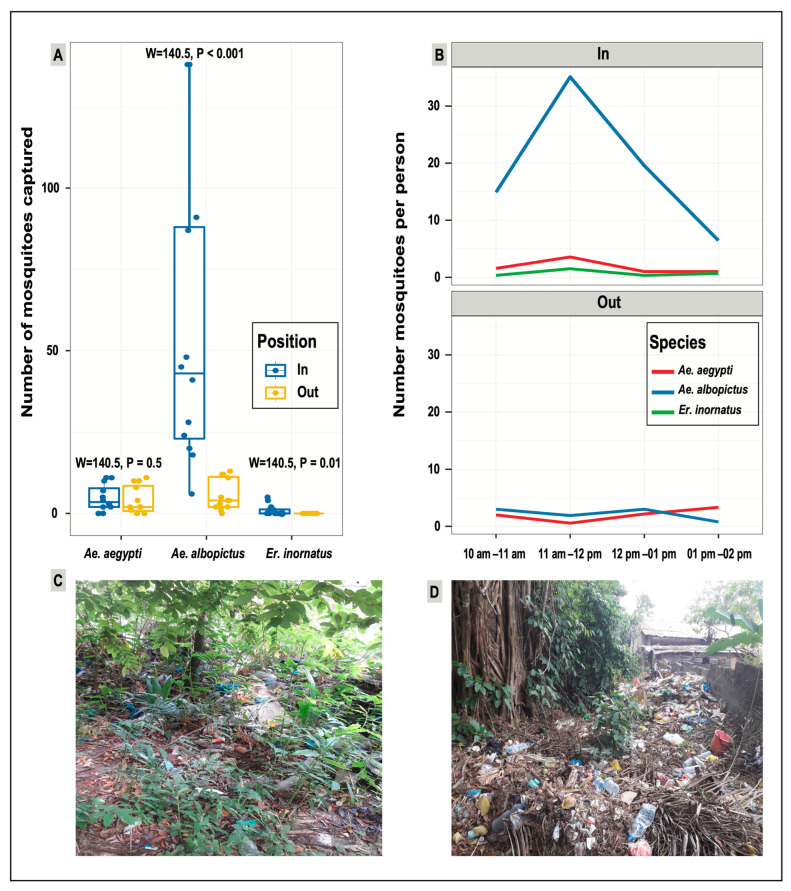
Number and biting rate of human-baiting mosquitoes in the Sibang arboretum and its surroundings. (**A**): Mean number of captured female mosquitos per person according to the spatial location. (**B**): Number of captured female mosquitoes per person over time inside and outside the forest. (**C**,**D**): A view of waste accumulating in the forest and its surroundings, illustrating the human footprints.

**Table 1 ijerph-20-05774-t001:** Water containers explored in the study area.

Habitat Nature	Inside the Forest	Outside the Forest	Overall
	(n = 28)	(n = 76)	(n = 104)
Artificial			
Glass jars	1 (5.0%)	1 (1.4%)	2 (2.1%)
Gutters	1 (5.0%)	16 (21.6%)	17 (18.1%)
Metallic containers	0 (0.0%)	6 (8.1%)	6 (6.4%)
Plastic containers	15 (75.0%)	17 (23.0%)	32 (34.0%)
Puddles	0 (0.0%)	18 (24.3%)	18 (19.1%)
Tires	0 (0.0%)	14 (18.9%)	14 (14.9%)
Wash basins	1 (5.0%)	1 (1.4%)	2 (2.1%)
Discarded freezer	0 (0.0%)	1 (1.4%)	1 (1.1%)
Discarded toilet bowls	2 (10.0%)	0 (0.0%)	2 (2.1%)
Subtotal	20 (100%)	74 (100%)	94 (100%)
Natural			
Puddles	1 (12.5%)	0 (0.0%)	1 (10.0%)
Tree holes	5 (62.5%)	0 (0.0%)	5 (50.0%)
Streams	2 (25.0%)	2 (100%)	4 (40.0%)
Subtotal	8 (100%)	2 (100%)	10 (100%)

n: number of water containers.

**Table 2 ijerph-20-05774-t002:** Assemblage of mosquitoes according to the breeding site type and spatial location.

Species	Inside Forest			Outside Forest			Overall
	Artificial	Natural	Sub-Total	Artificial	Natural	Sub-Total	
*Ae. aegypti*	0 (0.0%)	0 (0.0%)	0 (0.0%)	27 (4.3%)	0 (0.0%)	27 (4.1%)	27 (3.5%)
*Ae. albopictus*	67 (77.0%)	3 (13.6%)	70 (64.3%)	188 (29.7%)	0 (0.0%)	188 (28.5%)	258 (33.5%)
*An. gambiae* s. l.	0 (0.0%)	1 (4.6%)	1 (0.9%)	41 (6.5%)	0 (0.0%)	41 (6.2%)	42 (5.4%)
*Cx. cinerellus*	0 (0.0%)	0 (0.0%)	0 (0.0%)	1 (0.1%)	0 (0.0%)	1 (0.1%)	1 (0.1%)
*Cx. decens*	2 (2.3%)	0 (0.0%)	2 (1.8%)	0 (0.0%)	0 (0.0%)	0 (0.0%)	2 (0.3%)
*Cx. duttoni*	0 (0.0%)	0 (0.0%)	0 (0.0%)	6 (0.9%)	0 (0.0%)	6 (0.9%)	6 (0.8%)
*Cx. quinquefasciatus*	12 (13.8%)	0 (0.0%)	12 (11.0%)	222 (35.1%)	0 (0.0%)	222 (33.6%)	234 (30.4%)
*Cx. tauffliebi*	0 (0.0%)	11 (50.0%)	11 (10.1%)	17 (2.7%)	19 (67.8%)	36 (5.4%)	47 (6.1%)
*Cx. trifilatus*	0 (0.0%)	0 (0.0%)	0 (0.0%)	3 (0.5%)	0 (0.0%)	3 (0.5%)	3 (0.4%)
*Cx. umbripes*	0 (0.0%)	0 (0.0%)	0 (0.0%)	1 (0.1%)	0 (0.0%)	1 (0.1%)	1 (0.1%)
*Cx. univittatus*	0 (0.0%)	0 (0.0%)	0 (0.0%)	4 (0.6%)	0 (0.0%)	4 (0.6%)	4 (0.5%)
*Culex sp.*	0 (0.0%)	0 (0.0%)	0 (0.0%)	12 (1.9%)	1 (3.6%)	13 (2.0%)	13 (1.7%)
*Er. inornatus*	0 (0.0%)	0 (0.0%)	0 (0.0%)	3 (0.5%)	0 (0.0 %)	3 (0.5%)	3 (0.4%)
*Lu. tigripes*	6 (6.9%)	5 (22.7%)	11 (10.1%)	108 (17.1%)	8 (28.6%)	116 (17.5%)	127 (16.5%)
*Tx. evansae*	0 (0.0%)	2 (9.1%)	2 (1.8%)	0 (0.0%)	0 (0.0%)	0 (0.0%)	2 (0.3%)
Total	87 (100%)	22 (100%)	109 (100%)	633 (100%)	28 (100%)	661 (100%)	770 (100%)

## Data Availability

The data presented in this study are available on request from the corresponding author.

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
