# Peer review of "Urban Green Spaces and Vector-Borne Disease Risk in Africa: The Case of an Unclean Forested Park in Libreville (Gabon, Central Africa)"

_ijerph, 2023, doi:10.3390/ijerph20105774_

Round 1

Reviewer 1 Report

In the presented manuscript, the authors investigate the differences in mosquito populations between Sibang aboretum and the surrounding urban areas of Libreville, Gabon. The paper is reasonably well-written and as far as I can tell, the methodology is relatively straight-forward and free of fatal flaws. My major point of critique with this work is that the underlying data is only "available upon request". In the interest of science, it should be publicly archived and made available for meta analyses. In particular, species occurrence records should be submitted to the Global Biodiversity Information Facility (https://www.gbif.org/).

On a general note, the manuscript could benefit from a clear set of hypotheses or research questions that are stated in the beginning and referred back to in the Conclusions.

It also feel that the major take-away from this study is somewhat glossed over: While forested areas certainly seem to affect species composition etc., by far the most important factor locally seems to be trash and other anthropogenic containers. At least that is my impression after reading the results - should be discussed more thoroughly

some more specific comments:

page 1, line 41-43: "poor environmental practices" mentioned twice in the same sentence. But more importantly, how can urban forested ecosystems be considered "drivers" when the reason they "might boost urban mosquito densities" is "due to poor environmental practices"? Doesn't that mean that the actual driver and underlying problem is that people have no better place to get rid of their trash?

page 2, line 60: Why would a lack of water proliferate breeding sites?

page 13, line 467: Araujo reference is incomplete (probably [68]?)
page 13, line 468: I suppose that is about the temperature dependence of the extrinsic incubation period? Please add a reference.

Figure 1: In panel C, it appears that the locations of the pie charts have been adjusted so that they do not overlap. That makes sense, but should be mentioned in the description.

Reviewer 2 Report

Dear authors,

This is an interesting faunistic field study that gives information about the mosquito species composition and abundance in a green area of poor sanitary and the surroundings in an area of Africa.

Herein, you will find some comments of major importance that may benefit the manuscript.

The title and in many parts of the manuscript it is implied that the green space in the urban area may carry high mosquito populations of medical importance that may infest the urban area. However, it should be highlighted that the vegetated area is a poorly sanitated and this is the main reason of harboring mosquitoes, not just the green. From the results it is clear that mosquito abundance inside the green area was favored by the waste (poor hygiene conditions) there. Noteworthy, the dominant breeding sites in the forest were artificial man-made containers and the dominant species was Ae. albopictus which is a container breeding mosquito. So, in this perspective the following major revisions are required:

In the title: Change the title addressing the aforementioned, e.g., “Unclean” or “Poor hygiene conditions in…” urban green spaces and vector-borne disease risk in Africa: case of the Sibang forested park in Libreville (Gabon, central Africa).

Line 42: change to “…considering poor hygiene conditions in urban forested ecosystems…”

Line 325: change to “…how such a green area full of waste could…”

Line 463: change to “…the issue of the vegetation with poor sanitation of urban cities…”

Line 469: change to “… unclean urban forests…”

Lines 472-477: Revise these lines to highlight the need of high sanitation measures in green spaces in urban areas.

Line 141: Human bait method was applied only during the daytime, hence the collected mosquitoes were only day-time biting mosquito species like Ae. albopictus. This is a limitation of the study to be addressed in the discussion, since human bait method in the current study provided information only for day-time biting mosquito fauna.

“Larval prospectives” mean “larval samplings” and therefore this phrase should be changed in the manuscript accordingly.

The text of the abstract and the whole manuscript should be limited.

Reviewer 3 Report

This study addresses an important issue: assessing the risk of vector-borne diseases associated with green spaces. The concept of green spaces in urban areas has gained in popularity in recent decades, as they can improve human well-being by e.g. mitigating the urban heat island effect and by providing recreational spaces. However, green spaces could also support mosquito populations, which was studied in a preserved forest patch in the heart of the capital of Gabon, central Africa.

The study was conducted during the dry season (over a period of approx. 1.5 months) as this forest patch may serve as a refuge for mosquitoes during this season. This is also the main worry I have: This paper presents a snapshot of mosquito abundance and species diversity during a 1.5-month period in 2020. Yet the authors compare the results obtained from inside and outside the forest patch on multiple occasions (larval habitat comparison, mosquito abundance, species diversity) without making this limitation sufficiently clear throughout the text (title, abstract, discussion).

In addition to the time component, the authors state that “Mosquito breeding sites were prospected as exhaustively as possible depending on field accessibility and permission from residents to visit their properties”. Thus, larval sampling was not done systematically. I admit that this is nearly impossible to achieve in urban environments where we have to rely on the cooperation of the general public, but this does mean that important habitats (and thus a number of mosquitoes and potentially mosquito species) may be missed. While intuitively I do not think this will change your conclusion about the lower species diversity inside the forest patch, we cannot be certain.

Regarding the human landing catches, it seems to me that those catches were not conducted simultaneously inside and outside the forest patch (randomized). If one day the catches are done outside, and the other day inside the patch, there could be an effect of climatic conditions (temperature, humidity, rainfall, air pressure) or human movement (e.g. busier days, special events), which can differ between days and affect mosquito activity patterns. I am just making these up, simply to illustrate that there may be confounding factors that can affect your results, but are now not taken into account because of the chosen sampling strategy.

Minor comments

“(…) there is a high level of anthropogenic pressure from neighboring households on the arboretum ecosystem, which is characterized by an excessive accumulation of domestic waste inside the forest, and prone to favor the proliferation of major disease vector species, especially those exploiting artificial breeding containers such as Ae. albopictus, or Ae. aegypti or Cx. quinquefasciatus.” This (accumulation of waste) is used as an explanatory factor, but seems not to be studied systematically. Some images are provided, but it is not clear if artificial breeding sites inside the forest patch are mostly found in these waste areas, and if these areas are present all around the forest patch.

“larvae and pupae were placed into labeled trays covered with a mosquito net and maintained at room temperature until the emergence of adults”. Is there any information on mortality during the immature stages? I imagine immatures were reared in water that was collected from each breeding site? Just wondering if certain species do not develop properly under laboratory conditions, which may affect your species abundance (and maybe diversity) results.

Round 2

Reviewer 2 Report

I would like to thank the authors for their responses.

However, the following limitation of the study has not been appropriately addressed in the discussion:

Human bait method was applied only during the daytime, hence the collected mosquitoes were only day-time biting mosquito species like Ae. albopictus. This is a limitation of the study to be addressed in the discussion, since human bait method in the current study provided information only for day-time biting mosquito fauna.

R: We thank the reviewer for the remark. Human landing catches could have been performed during a longer period of time, including daytime and night. Unfortunately, due to general restriction measures related to COVID-19, we could not perform extended captures at that time.

Reviewer 3 Report

I am satisfied with the authors' responses to my questions/issues raised in my initial review. The manuscript is revised thoroughly, which is much appreciated!
